# From Structural Studies to HCV Vaccine Design

**DOI:** 10.3390/v13050833

**Published:** 2021-05-04

**Authors:** Itai Yechezkel, Mansun Law, Netanel Tzarum

**Affiliations:** 1Department of Biological Chemistry, Alexander Silberman Institute of Life Sciences, Faculty of Mathematics & Science, The Hebrew University of Jerusalem, Jerusalem 9190401, Israel; itai.yechezkel@mail.huji.ac.il; 2Department of Immunology and Microbiology, The Scripps Research Institute, La Jolla, CA 92037, USA

**Keywords:** hepatitis C virus (HCV), neutralizing antibodies, structural studies, envelope glycoproteins, E1, E2, E1E2 complex, V_H_1-69, neutralization face, vaccine design

## Abstract

Hepatitis C virus (HCV) is a serious and growing public health problem despite recent developments of antiviral therapeutics. To achieve global elimination of HCV, an effective cross-genotype vaccine is needed. The failure of previous vaccination trials to elicit an effective cross-reactive immune response demands better vaccine antigens to induce a potent cross-neutralizing response to improve vaccine efficacy. HCV E1 and E2 envelope (Env) glycoproteins are the main targets for neutralizing antibodies (nAbs), which aid in HCV clearance and protection. Therefore, a molecular-level understanding of the nAb responses against HCV is imperative for the rational design of cross-genotype vaccine antigens. Here we summarize the recent advances in structural studies of HCV Env and Env-nAb complexes and how they improve our understanding of immune recognition of HCV. We review the structural data defining HCV neutralization epitopes and conformational plasticity of the Env proteins, and the knowledge applicable to rational vaccine design.

## 1. Introduction

Viral hepatitis was first described by Hippocrates in approximately 400 BC. However, the causative agents were only discovered in the second half of the twentieth century. Most viral hepatitis cases are caused by one of the five unrelated hepatotropic viruses, hepatitis A–E, where hepatitis B and C are responsible for more than 95% of the mortality cases [1]. Discovery of hepatitis viruses and consequently the development of vaccines against hepatitis A, B, and E contribute to the controlling of viral spread. Unfortunately, no vaccine is currently available for hepatitis C virus (HCV).

HCV is a bloodborne virus commonly transmitted by transfusion of unscreened blood and blood products, unsafe healthcare practices, sharing needles between people who inject drugs (PWIDs), and contaminated equipment in tattoo parlors. According to the last World Health Organization (WHO) global hepatitis report [1], 71 million people were infected by HCV worldwide in 2015, ~1% of the world’s population. HCV causes ~500,000 deaths and ~2 million new infections annually [1,2,3]. Approximately 25% of acute HCV infection results in spontaneous viral clearance, usually within the first 12 months of infection. The remainder develop a chronic hepatitis C (CHC) infection that can lead to liver cirrhosis (in ~20% of the cases) and, eventually, hepatocellular carcinoma [4,5].

In the last decade, HCV treatment has substantially changed with the clinical implementation of direct-acting antivirals (DAA) that target HCV nonstructural (NS) proteins crucial for viral replication. Since 2014, a second generation of the DAAs have become available with a cure rate of better than 95%. Nonetheless, DAA treatment faces several challenges: (a) HCV infection can remain asymptomatic for years [6], and during this time many infections go undiagnosed while patients suffer from sustained liver damage; (b) DAA treatments do not prevent reinfections [7]; (c) DAA-resistant viruses can emerge; (d) patients with advanced liver disease remain at risk of liver cancer; and (e) HCV is highly prevalent in developing countries and among marginalized populations where access to HCV diagnosis and treatment is limited [8]. Indeed, many at-risk groups (e.g., PWIDs) spread the infection faster than they are being cured. These challenges highlight the critical need of a prophylactic vaccine for HCV eradication [6,9,10].

HCV is an enveloped positive-sense single-strand RNA virus classified within the Hepacivirus genus, one of the four genera of the Flaviviridae family. The HCV positive-sense, single-stranded RNA genome encodes a single polyprotein that is processed by host and viral proteases into three structural proteins (core, E1, and E2) and seven NS proteins (p7, NS2, NS3, NS4A, NS4B, NS5A, and NS5B). HCV has high genetic diversity with six major and two minor genotypes (genotypes 1–8), and 90 subtypes [9]. In addition, NS5B, HCV RNA-dependent RNA polymerase lacks proofreading activity, giving rise to the heterogeneous viral quasispecies within infected individuals and immune escape [10].

The extreme genetic diversity of HCV is a major roadblock for vaccine development. Nevertheless, the spontaneous viral clearance suggests that chronic HCV infection is preventable, if a robust, broadly effective immune response can be induced by vaccination. Evidence from human and chimpanzee studies indicate that both B-cell and T-cell responses are associated with viral clearance (reviewed in [11,12]). In this context, HCV clearance is closely related to the eliciting of a strong and early neutralizing antibody (nAb) response that targets HCV Env glycoproteins [13,14].

Diverse strategies to induce humoral and/or cell-mediated immunity have been described [12,15,16,17,18,19], including viral vectors that express multiple HCV antigens [20,21,22], DNA vaccination [23], recombinant E2 and E1E2 protein vaccination [24,25,26,27], HCV viruslike particles (VLPs) [28,29], and, recently, antigen-displaying lipid-based nanoparticle vaccines [30] and self-assembly nanoparticles [31]. Nevertheless, at present, only three vaccine candidates were proceeded into human preclinical and clinical trials [19,32]. The first vaccine candidate is a prototype vaccine with the HCV core protein that was tested for its ability to induce T-cell responses in healthy individuals not at risk for HCV infection. However, as T-cell responses were detected in only 25% of the participants, the vaccine was not advanced further to at-risk populations [33]. The second vaccine candidate is an adenovirus-based vector that expresses the NS proteins and aims to trigger an HCV-specific T-cell response. Proof-of-concept vaccination experiments on the chimpanzee model resulted in the priming of cross-reactive HCV-specific T-cell response and the suppression of acute viremia [22]. A subsequent experiment in healthy human volunteers also showed the priming of a broad and cross-reactive HCV-specific T-cell response [22,23,34]. These results led to a randomized, double-blind, placebo-controlled phase 2 clinical trials in high-risk PWIDs (NCT01436357), but unfortunately, the vaccine candidate failed to induce protective effects against chronic HCV infection [35,36].

The third vaccine candidate is a recombinant form of HCV full-length E1E2 envelope (Env) glycoproteins from the genotype 1a strain HCV-1 (Chiron vaccine). Animal vaccination experiments resulted in effective immunogenicity and good antibody response against homologous or heterologous HCV [34,37,38]. A subsequent phase 1, placebo-controlled, clinical trial (NCT00500747) of the Chiron vaccine resulted in the induction of nAbs and proliferative T-cell responses against E1E2 Env in healthy human volunteers [26,27,39]. However, cross-nAb responses were only elicited in a few subjects, and a substantial portion of the nAb response was isolate-specific, directed toward variable regions (VR) of the Env proteins [25,40].

The results of the clinical trials suggested the need to design Env-derived antigens for the induction of a high-titer cross-nAb response in future vaccine candidates. A molecular-level understanding of the nAb responses against HCV is imperative for targeted engineering and rational vaccine design. In this context, characterization of the interactions between broadly nAbs (bnAbs) and HCV Env can provide critical information about conserved epitopes on the viral Env proteins. During the last two decades, bnAbs to the HCV Env proteins were isolated from infected patients and immunized animals [41,42]. Although nAbs that target E1 (e.g., IGH526 and IGH505 [39]) or bnAbs that target conformational E1E2 epitopes (e.g., AR4A and AR5A) were reported [43], the majority of them target E2. Epitope-mapping experiments indicate that the majority of the cross-neutralizing epitopes overlap with the CD81 receptor-binding site, suggesting that the main neutralization mechanism is blockage of virus interaction with this host receptor (32). Cross-neutralizing epitopes that overlap with the CD81 receptor-binding site (e.g., AR3, AS412, and AS434; see below) are highly conserved among HCV genotypes [44,45]. In this review, we summarize the recent advances in structural studies of HCV Env and how this knowledge can be applied to support rational vaccine design.

## 2. HCV Env Glycoproteins

HCV particles consist of a nucleocapsid containing the viral genome, surrounded by an endoplasmic-reticulum-derived membrane crowned by the E1 and E2 Env proteins [46]. A hallmark of the HCV particle isolated from infected plasma is its appearance as a lipoviral particle (LVP) in association with host-derived apolipoproteins [47,48] that may reduce virus sensitivity to the neutralizing antibody response [49,50,51,52]. HCV Env mediates viral entry into the host hepatocytes [53], a process that involves interactions with a plethora of host factors [54]. HCV particles are initially captured by heparan sulfate proteoglycans (HSPG) and the low-density lipoprotein receptor (LDLR), followed by a high-affinity interaction with the scavenger receptor class B member 1 (SR-BI). The interaction with SR-B1 is most likely followed by the interactions of viral particles to the HCV main receptor, the tetraspanin CD81, and the activation of signaling pathways [54]. Consequently, virus and CD81 diffuse toward the tight junction and interact with the tight-junction proteins CLDN1 and OCLN to form the receptor clustering that allows internalization of the viral particle through clathrin-dependent endocytosis [54].

### 2.1. E1E2 Heterodimer

E1 and E2 are type I transmembrane glycoproteins with a C-terminal transmembrane domain (TMD) that form a heterodimer on the viral envelope [55]. The TMDs anchor E1 and E2 to the membrane and play a major role in the heterodimerization and assembly of the native E1E2 complex [56]. Epitope-mapping experiments of the AR4A and AR5A bnAbs, which target conformational E1E2 epitopes, suggested that the N terminus of E1 is associated with the C terminus of the E2 ectodomain [43]. HCV Env proteins are heavily modified post-translationally (see below), which challenges structural studies of HCV Env, and consequently, structure-based vaccine design. To date, there is no high-resolution structure of the E1E2 heterodimer, or the entire E1 or E2. Moreover, correct folding of E1 depends on E2 [57]. So far, only the ectodomain of E2 can be expressed as a folded and soluble monomeric protein and thus is amenable for structural studies [58,59,60].

#### Structural Studies of E1E2

A major obstruction of structural studies of E1E2 heterodimer is the inability to produce recombinant soluble E1E2 heterodimers. The deletion of the TM domains abolished the formation of natively folded E1E2 heterodimer and resulted in poor-quality protein samples [56]. Consequently, different approaches have been attempted to design soluble E1E2, including covalent linkage of E1 and E2 ectodomains using linkers [61,62] or fusion to Fc dimerization domains [26]. However, these approaches did not improve the expression level or heterogeneity compared to the unmodified E1E2.

Recently, two studies reported the design of soluble, scaffolded E1E2 heterodimers. Scaffolding Env proteins by attachment of C-terminal oligomerization domains, e.g., foldon, to facilitate nativelike assembly of the Env proteins was previously applied to several viruses, including respiratory syncytial virus (RSV), influenza hemagglutinin, and severe acute respiratory syndrome coronavirus (SARS) (for a review, see [63]). In the first study, Cao et al. expressed E1 and E2 fused to either an IgG Fc tag or a de novo designed heterodimeric tag (DHD15, PDB entry: 6DMA) in order to hold E1 and E2 in close proximity to increase the likelihood of forming the desired dimerization [64]. The fused proteins were expressed as soluble proteins, but with some heterogeneity of the oligomerization state. Structural characterization of the E1E2 heterodimer by electron microscopy together with a coevolution-based modeling strategy indicated potential interactions between E1 pFP, E2 VR2, and the back layer. In addition, the scaffolded proteins showed binding to E1-specific and E2-specific conformational mAbs and the CD81 receptor. Nevertheless, it has not been shown to bind to conformational E1E2 bnAbs, suggesting that this scaffolded E1E2 may not represent a native heterodimeric assembly. In the second study, E1 and E2 ectodomains were expressed as cleavable polyproteins fused to a self-assembling heterodimeric, homotrimeric, or heterohexameric scaffold. One of them, E1E2.LZ, used the human c-Fos/c-Jun leucine zipper heterodimer to scaffold E1 and E2. Soluble E1E2.LZ showed high solubility and low heterogenicity, and also bound to a broad panel of conformational E2 and E1E2 antibodies, as well as the CD81 receptor, and displayed immunogenicity of native E1E2 Env in mice, providing evidence for the assembly of a native-like soluble E1E2 heterodimer. Such scaffolded E1E2s can potentially be used for structural studies and as a platform for E1E2-based vaccine design.

### 2.2. E1 Env

E1 (amino acids 192–383 in the H77 prototypic strain UniProt number P27958, Figure 1) is heavily modified post-translationally. E1 contains four highly conserved N-linked glycans and eight cysteines (a fifth glycosylation site is present at position 250 in genotypes 1b and 6, or at position 299 in genotype 2b). Several studies have been conducted on the E1 disulfide bond network with conflicting conclusions as to whether disulfide bridges were formed intramolecularly and intermolecularly [65,66,67]. Mutagenesis experiments of E1 cysteines indicated that they have little effect on virus infectivity [68]. The E1 central region contains a conserved region that has been proposed to be a putative fusion loop (pFP) for fusion of viral envelope with the host membrane. The pFP is followed by a highly conserved region (CR), and its function is poorly characterized [69]. The N-terminal domain (NTD) of E1 contains the majority of post-translational modifications, suggesting it is an exposed domain on the virus surface [66]. The role of E1 in HCV entry and immune evasion is still not understood, but it was suggested that E1 may contribute to the attachment of HCV to hepatocytes by interactions with apolipoproteins on the lipoviro particles, or by direct interaction with the scavenger receptor class B member 3 (SCARB3 or CD36) [66,70,71,72]. The immunogenicity of E1 is relatively low; however, two neutralization epitopes were defined: the E1 N terminus that is targeted by the weakly nAb H-111 [73]; and the CR, targeted by the weakly nAb IGH526 and the non-nAb IGH505 [39].

#### Structural Studies of E1 Env

Structural studies of E1 are limited to partial structures of the NTD [67], CR C-terminus [74,75], and NMR structure of the TMD (Figure 1) [56]. The NTD crystal structure (consisting of amino acids 192–270, [67]) shows a covalently linked, domain-swapped homodimer with a well-defined secondary structure. However, this structure was obtained in low pH conditions and therefore may represent the postfusion conformation and not the folding of E1 that is presented on the viral Env. The structure of the E1 CR C-terminus was solved by NMR (amino acids 314–342) [75] and X-ray crystallography in complex with the IGH526 mAb [74], revealing that amino acids 314–324 adopt a helical conformation. Nonetheless, molecular dynamic simulations suggested that the unliganded epitope is flexible in solution [74].

### 2.3. E2 Env

E2 (amino acids 384–746, Figure 1) is the HCV receptor-binding protein and, consequently, the main target for bnAbs. The E2 ectodomain is heavily modified by nine strictly conserved disulfide bonds that contribute to its unexpected high thermal stability (melting temperature of ~85 °C, [76]). E2 is also modified by up to 11 highly conserved N-linked glycans (Figure 1) that reduce HCV immunogenicity by shielding conserved epitopes from the neutralizing immune response [77,78]. The E2 ectodomain consists of an N-terminal hypervariable region 1 (HVR1), a central core domain, and a C-terminal stalk region that connects the E2 ectodomain to the TM region (Figure 1). The E2 core domain (E2c) adopts a globular structure that consists of a central immunoglobulin (Ig) β-sandwich fold, which is stabilized by disulfide bonds and flanked by an N-terminal front layer and a C-terminal back layer (Figure 2a). A long loop, named the CD81 binding loop, extends from the central β-sandwich domain and contains a critical region for CD81 receptor binding. An EM study of the H77 E2 ectodomain-AR3A bnAb complex indicated that the ectodomain also adopts a globular structure. The HVR1 region is situated in front of the front layer and the stalk is situated behind the back layer and VR3 [44]. E2 possesses three variable regions: HVR1 and variable regions 2 and 3 (VR2 and VR3) (Figure 1), which increase the genetic diversity of HCV to efficiently evade nAbs. It was recently suggested that a closed conformation of HCV Env is stabilized by the HVR1 and protective glycans to regulate HCV neutralization and interactions with the SR-B1 receptor [79].

#### 2.3.1. Structural Studies of E2 Env

Beside the post-translational modifications, E2 exhibits high intrinsic flexibility. Structural studies of E2 indicated that more than 60% of E2 ectodomain residues are disordered or in loops [44,80]. Hydrogen–deuterium exchange (HDX) mass spectrometry experiments [74,80] and molecular dynamics (MD) simulation [44,81] indicated that E2 variable regions and the receptor-binding site exhibit high conformational flexibility. Despite these challenges, the advance in structural studies of E2 Env during the last decade contribute to the characterization of the E2 Env structure and the antigenic sites for understanding of antibody recognition of HCV. This knowledge provides a useful molecular template to enable structure-based design of candidate vaccine antigens. To date, three overlapping neutralizing sites (antigenic site 412 (AS412), antigenic site 434 (AS434), and antigenic region 3 (AR3)) and a linear epitope for non-nAbs binding (a.a. 529–540) have been structurally identified. The neutralizing sites are composed of highly conserved residues across HCV genotypes and cluster on an exposed surface on E2 Env overlapping with the CD81 receptor-binding site.

##### AS412

The initial structural studies of E2 were focused on crystal structures of bnAb Fab fragments in complex with epitope-derived peptides from two neutralization sites, AS412 (comprising residues 412–423) and AS434 (residues 434–446) (reviewed in detailed in [82,83,84]) (Figure 2b). AS412 (also called antigenic domain E, or epitope I) is a highly conserved linear antigenic site that connects the HVR1 and the front layer. It contains the first two N-glycosylation sites of E2, N417, and N423 (Figure 1), and residues that are critical to CD81 receptor binding (e.g., W420 [45]). Therefore, AS412 is the target for some well-characterized bnAbs, but also a hotspot for antibody evasion through shifting of the glycan shield [25,85,86,87,88,89,90,91]. Natural elicitation of such bnAbs in HCV-infected patients is relatively rare and found in 2–15% of the cases [92,93,94].

Crystal structures of AS412 in complex with 11 different human and rodent nAbs (Table 1) demonstrated that AS412 can adopt at least three conformations for neutralization: a β-hairpin, an open, and a semiopen conformation. The β-hairpin conformation is the first to be determined and the most common one, as observed in the complexes with HCV1, AP33, MRCT10.v362, hu5B3.v3, MAb24, 19B3, and 22D11 nAbs [90,92,93,94,95,96,97]. The β-hairpin conformation is stabilized by a number of internal backbone hydrogen bonds and contains a hydrophobic face recognized by the antibody heavy and light chains (HC and LC). N417 and N423 glycosylation sites are projected from the opposite side of the peptide and are solvent-exposed [94], indicating that AS412 is likely not closely packed against the E2 core domain. The semiopen conformation was observed in the complex structures with human bnAbs HC33.1, HC33.4, and HC33.8 [98,99]. In this conformation, the majority of the epitope (excluding residues 414 and 415) adopts an extended conformation that is stabilized by one internal backbone hydrogen bond [98]. The extended open conformation was observed in the crystal structure with rat nAb 3/11 [100], which is stabilized by internal backbone hydrogen bonds. Despite the differences in the AS412 conformations and the mode of binding by the nAbs, alanine scanning mutagenesis and structural analysis indicate that the highly conserved L413, G418, and W420 are critical for the binding of AS412 bnAbs (beside 3/11 [101]).

The different structures of AS412 likely reflect an intrinsic conformational variability of this E2 region, and suggest, as confirmed by recent MD studies [81,107], that AS412 is dynamically sampling a range of defined conformations (at least three). This flexibility may explain the modest immunogenicity of AS412. A second level of flexibility is the position of the AS412 region relative to the E2 core domain. This flexibility was validated by an EM study on the H77 E2 core–HCV1 bnAb complex, which revealed a 10–22° variation in the angle that the AS412-binding Fab fragment approaches E2 [76].

HCV applies multiple evasion mechanisms in order to evade the immune system, including high genetic variability, glycan shielding, conformational flexibility, immune decoy epitopes, and the occurrence of escape mutations (reviewed in [16,42]). Most of the escape mutations occur within nAb-binding epitopes; nevertheless, mutation in regions that are not directly involved in the epitopes, but reduce the efficiency of nAbs, can also result in viral escape [108,109,110]. Escape mutations in AS412 have been reported in several studies [90,96,111,112,113,114], including the N415D/K/Y and N417S/T mutations. The N417S/T mutations can cause a glycosylation shift from N417 to N415. Structural analysis of AS412–nAb complexes indicates that in the case of the β-hairpin and the open conformations, the side chain of N415 is buried in the antibody-binding pocket. Therefore, the N417S/T mutation and the subsequent glycosylation shift to N415 would create steric clashes in the antibody-binding pocket and interfere with antibody binding [105,112,113]. In contrast, the neutralization potency of the HC33 bnAbs that bind the semiopen conformation is not impaired by the glycosylation shift to N415 because the side chain of N415 is solvent-exposed in the complex structure [92,103,115].

##### AS434

AS434 (also called antigenic domain D or epitope II) is a highly conserved epitope among HCV genotypes that contains residues involved in the binding of CD81 receptor (Figure 2b) [115,116,117]. Several human bnAbs (HC84s) [110] and weakly neutralizing murine Abs (mAb#12 and mAb#8) [104,105] that target AS434 have been isolated (Table 1). In vitro selection of escape mutants using HC84 bnAbs did not result in escape virus [110], indicating the AS434 is a good target for structure-based vaccine design.

Structural analysis of AS434–mAb complexes indicated that AS434 is a short 1.5-turn α-helix (residues 437–442) flanked by an N- and a C-terminal extended region [102,103]. For human bnAbs HC84.1, HC84.27, HC84.26, and HC84.26AM, the interactions are dominated by hydrophobic interactions between the side chains of the α-helical residues L441 and F442 and the antibody complementarity-determining regions (CDRs) [102,110]. In contrast, for mouse mAb#12 and mAb#8, the interactions are dominated by hydrophilic interactions with the side chains of E431 and N434 and hydrophobic interactions with the side chains of W437 and L438. In the crystal structures of the E2 core [44,80,118,119,120,121], W437 and L438 are solvent-inaccessible, thus indicating that neutralization by mAb#12 and mAb#8 requires conformational rearrangement of AS434 to expose the side chains of W437 and L438.

##### Amino Acids 529–540

A third linear epitope for mAb binding, spanning residues 529–540 (epitope III), was structurally characterized in the crystal structure of DAO5 in complex with epitope-derived peptides. DAO5 is a non-neutralizing mAb that binds an epitope composed of the C-terminus of CD81 binding loop and the β-strand 6 of the β-sandwich scaffold region [106]. Crystal structures of DAO5–peptide complexes indicate that the peptide adopts a one-turn α-helical conformation (residues 534–539) with the side chains of F/M537 and L539 buried in the antibody-binding interface. In contrast, in the crystal structures of E2 in complex with bnAbs [44,80,118,119,120,121], this region adopts a β-strand conformation, while F537 and L539 are solvent-inaccessible.

##### E2 Core Structures

Since 2013, structural information on E2 Env has been accumulated. Extensive engineering of E2 ectodomain enabled the first structural determination by X-ray crystallography of the core domain of E2 in two independent studies [44,80]. In both cases, a bound mAb facilitated crystallization of E2. The first structure, derived from the H77 isolate (genotype 1a) in complex with bnAb AR3C that targets antigenic region 3 (AR3), consists of E2 residues 412–645 with an internal truncation of VR2 and removal of the N448 and N576 glycosylation sites (E2c, Figure 1) [44]. The second structure, the J6 isolate (genotype 2a) in complex with non-nAb 2A12, consists of E2 residues 456–656 (456–652 in H77 numbering; named core E2, Figure 1) [80]. Both structures revealed a novel protein fold of the E2 core domain with an overall similar fold but with some variations in the respective disulfide bond network (recently reviewed in [122]).

Structural analysis of the H77 E2c–AR3C complex indicated that AR3 is a cluster of discontinuous epitopes that overlap a hydrophobic surface formed by the E2 front layer and CD81 binding-loop regions [47,122,123] (Figure 2a). Originally, AR3 was defined by a panel of potent human bnAbs (AR3A-D), isolated from a chronic HCV patient [43,124]. AR3 bnAbs could abrogate an ongoing HCV infection in genetically humanized mouse models and protect animals from HCV challenge in passive antibody transfer experiments [46,123,125]. Now we know that the antigenic surface defined by AR3 covers epitopes targeted by bnAbs that bind conformational epitopes overlapped with the E2 front layer and CD81 binding loop (AR3-specific or AR3-targeting bnAb), and were isolated from different HCV patients [123,126,127,128,129,130] and immunized animals [125]. AR3 comprises mostly highly conserved residues across HCV genotypes [44], including residues that are critical for CD81 binding. Recent virus escape studies using the AR3A bnAb demonstrated a high barrier to resistance, suggesting that AR3 is an optimal target for rational vaccine development [131].

Based on the H77 E2c structure and epitope-mapping experiments, an accessible antigenic surface on HCV Env, the E2 neutralizing face, was defined [44]. Of note, the neutralizing face is not modelled in the J6 E2 structure [80]. The neutralizing face is a predominantly hydrophobic surface that overlaps with the E2 main neutralization sites: AS412, AS434, and AR3 (Figure 2b). The neutralization face is surrounded by N-glycosylation sites (N423, N430, N532, and N623), yet it is not obstructed by glycans as suggested by negative-stain EM structure (excluding the AS412 region) [44]. Moreover, EM studies indicated that the neutralization face can be recognized by nAbs with different angles of approach [76].

From the publication of the second E2 structure (early 2014) to 2018, structural knowledge was limited to these two E2 core structures, underscoring the difficulty of structural studies of HCV Env E2. At the end of 2018, the Bjorkman lab [118] published additional structures of the E2 ectodomain (without the stalk region, E2ecto; Figure 1) and E2 core of genotype 1 (isolates 1b09 and 1a53) in complex with two AR3-targeting bnAbs, HEPC3 and HEPC74, which were isolated from individuals who spontaneously cleared HCV infection [126]. To enhance crystallization, the E2 ectodomain was coexpressed with Fabs to assemble stable E2–Fab complexes, a method that was previously used to determine the structure of the respiratory syncytial virus (RSV) F glycoprotein [132]. Later, the Bjorkman lab also determined the structure of nAb AR3X (isolated from the same patient as for the AR3 bnAbs [124]) in complex with 1b09 E2ecto [120]. Shortly after, we published the structures of the modified E2 core (E2c3, E2c with VR3 deleted; Figure 1) from genotype 6a (isolate HK6a) in complex with the AR3A/B/D bnAbs [119]. In a follow-up study [121], we determined structures of E2c3 in complex with bnAbs that were isolated from chronically infected HCV patients (U1, HC11, and HC1AM [124,128]) and from individuals who spontaneously cleared HCV infection (212.1.1 [127]). Recently, we published the structure of HK6a E2c3 in complex with AR3-targeting bnAbs (RM2-01 and RM11-43) [133] isolated form rhesus macaques (RM) immunized with the Chiron E1E2 antigen [125]. In total, structures of E2 fragments in complex with 13 different bnAbs are now available for understanding how cross-neutralization of HCV is achieved by the humoral immune response against AR3.

#### 2.3.2. The Binding Mode of AR3-Targeting bnAbs

Genetic analyses of bnAbs targeting AR3 indicated that the HC variable region (IGHV or V_H_) is frequently encoded by the V_H_1-69 germline gene family [43,123,124,126,127,128,129]. The V_H_1-69 antibody gene family encodes an unusually hydrophobic CDR2 (CDRH2), and is frequently isolated in protective antibody responses against viruses; e.g., influenza A virus and HIV-1 (reviewed in [134]). Of note, HC84 bnAbs that target AS434, as well as some non-nAbs that target AR1, are also encoded by V_H_1-69 germline genes.

The first crystal structure of H77 E2c in complex with AR3C identified interesting features in the antibody, characterized by a hydrophobic CDRH2 tip that serves as a crucial anchor for interactions with a highly conserved hydrophobic pocket formed by the E2 front layer and the CD81 binding-loop regions (overlaps AR3), and a long CDRH3 loop (18 a.a. and up, Table 2). In addition, the interactions are mainly contributed by the antibody HC [44]. These features are also found in the recent structures of other V_H_1-69-derived, AR3-targeting bnAbs. The features include: (1) an unusually hydrophobic CDRH2 that interacts with different hydrophobic patches on AR3; (2) a longer than average CDRH3 loop of 15–22 a.a. (compared to 9–15 a.a. for healthy human donors [119,135]); (3) interactions mediated mainly by the HC; and (4) a relatively low level of somatic hypermutation (SHM) [44,118,119,120,121].

Despite these common binding features, structural analysis of the E2–bnAb complexes revealed different modes of E2 binding. This suggests the V_H_1-69-encoded AR3-targeting bnAbs can potentially be clustered into subgroups based on their genetic and structural characteristics (Figure 3a and Table 2). Of note, in many cases, bnAbs isolated from the same patients share a similar mode of binding [123,124,137]. On the other hand, AR3A-D and HC11, and HEPC3 and HEPC74, that were isolated from different patients also share a highly similar mode of binding [121]. All AR3-targeting bnAbs bind epitopes that are composed of the front layer and CD81 binding loop; however, variation in the angle of Fab approach to E2 results in notable differences in the HC CDRs footprint (Figure 3a) [125]. For example, the CDRH2 of HEPC3, HEPC74, AR3X, RM2-1, and RM11-43 interacts with a hydrophobic core that is formed by the C-terminus of the front layer (a.a. 439–443), while for the rest, the hydrophobic core is formed by all of the front layer and the tip of CD81 binding loop. In addition, the CDRH1 of RM2-1 and RM11-43 interacts with the CD81 binding loop, whereas the other bnAbs interact with the CD81 binding loop with the CDRH2 or CDRH3. Difference in the angle of approach and, to some extent, in CDRH3 length can also influence the dominancy of the HC in the interactions. For some bnAbs, e.g., AR3A/B/C/D and HEPC3/74, interactions that are critical for E2 binding are mediated by the HC (HC dependent; Table 2), whereas for some bnAbs, the interactions require both HC and LC (HC dominant; e.g., HC1AM and the nAb 212.1.1) [121,134].

Another feature that characterized a subgroup of AR3-targeting bnAbs is an intramolecular disulfide bond within the CDRH3 loop. Some human V_H_1-69-encoded bnAbs that contain long CDRH3 (more than 17 a.a.: AR3A, AR3C, AR3X, HC11, HEPC3, and HEPC74) utilize the germline D gene segment 2-15 or 2-21 (D_H_2-15 or D_H_2-21) that contains a disulfide motif (CxGGxC). The disulfide bond, together with intra-CDR hydrogen bonds, help to stabilize a β-hairpin conformation of their CDRH3 (Figure 3b and Table 2) [44,118,119,120,121]. However, the CDRH3 of AR3B and AR3D adopt a similar β-hairpin conformation that is stabilized by only intra-CDR hydrogen bonds, indicating different solutions can be invented by the immune system to shape the CDRH3 into the β-hairpin conformation required for antibody function [119]. Recently, Yi et al. [130] reported the isolation from a chronic HCV patient of the bnAb 8D6. 8D6 targets epitope that overlap AR3 and contains the CxGGxC motif, yet it is encoded by the V_H_7-4-1 and D_H_5-18 genes, suggesting that bnAbs with different genetic background can share similar binding features. Taken together, structural analysis of V_H_1-69-encoded AR3-targeting bnAbs illustrates the high plasticity of the AR3 region, which can accommodate interactions with different bnAbs that share similar features but vary in CDRs sequence and CDRH3 length.

#### 2.3.3. E2 Immunodominant Decoy Epitopes

Biochemical and structural characterization of immunodominant decoy epitopes can support vaccine design to suppress these epitopes (e.g., by truncation or modifying) with the goal to focus antibody response on neutralizing epitopes to prime a more effective immune response. Several studies indicated that HVR1 serves to protect HCV from nAbs by diverting the humoral immune response from more conserved epitopes (reviewed in [137]). These observations led to the design of several HVR1-deleted vaccine candidates [138,139,140] with conflicting results regarding the effect of truncation of HVR1 on antigen immunogenicity and neutralizing antibody response [137].

Another immunodominant decoy epitope that was recently characterized structurally is the β-sandwich loop (a.a. 541–549) that connects the β6 and β7 strands of the E2 core [43,127,141]. This loop contains the seventh N-glycosylation site (N540) that is highly conserved in genotypes 1, 2, 4, and 5, but not in 3 and 6 because of the N540E substitution (Figure 1). The tip of the β-sandwich loop contains residues that are part of the antigenic region 1 (AR1), a weakly/non-neutralizing epitope targeted by antibodies AR1A, AR1B, E1 (isolated from the same patient as for AR3 bnAbs [46,123,141]), and HEPC46 (isolated from the same patient as for HEPC74 bnAb [118]). Of note, AR1A and E1 are also encoded by the V_H_1-69 germline gene [43]. Structural analysis of E2 complexes indicated that the β6-β7 loop can adopt one of the two distinct conformations, suggesting that this region may have some inherent flexibility [121]. In addition, next-generation sequencing (NGS) analysis of the HCV-immune repertoire, from which the AR1- and AR3-antibodies were isolated, confirmed the dominance of AR1A-like mAbs in antibody response to HCV [119]. It is known that the β-sandwich loop can be truncated from E2 without affecting the overall folding of the antigen [136], and it would be interesting to study whether such truncation may shift the antibody response to E2-conserved neutralizing epitopes in vaccination.

#### 2.3.4. Structural Plasticity of E2 

The entry process of enveloped viruses into their host cells requires conformational change of the Env proteins to mediate fusion of the viral and the host cell membranes [63,142]. Structural knowledge on the different conformations of HCV Env and their roles in the transition between the different states in HCV entry will be critical for understanding the virus entry mechanism of Hepaciviruses. It was initially hypothesized that E2 have a class II fusion protein fold, similar to other members in the Flaviviridae family. However, structural determination of the first E2c structures [44,80] demonstrated that E2 acquires a different fold, thus the entry mechanism of Hepaciviruses is likely different.

In the E2 structures, the front-layer region is constrained by two disulfide bonds (C429-C503 and C452-C620), which covalently anchor the front layer to the β-sandwich and back layer. Superposition of all available E2 structures indicated that the front layer can adopt at least two conformations when bound to bnAbs. The “A” conformation is the first known and the most common conformation [44]. In the “A” conformation, the front layer adopts a helix-turn helix like shape that obscures large sections of the E2 β-sandwich and back-layer regions. The “B” conformation was recently discovered in the crystal structures of H77 E2c3–Fab 212.1.1 and H77 E2c3–Fab HC1AM complexes (Figure 4a). In these structures, the central part of the E2 front layer (amino acids 430 to 451) adopts an alternative conformation that exposes some sections of the E2 β-sandwich and back-layer regions (Figure 4b) [121]. As a result, several residues that are critical for the binding with the CD81 receptor (e.g., Y613 and W616 [117]) become accessible for direct interaction with the bnAbs. In addition, the CDRH2 loops bind more deeply into a new hydrophobic groove formed by the front layer, CD81 binding loop, and back layer. The existence of the B conformation, which involves full exposure of Y613 in the back-layer region, is supported by the report that in T cells, Y613 is exposed on E2 and can be phosphorylated by Lck tyrosine kinase for immune evasion [143].

Given that bnAbs recognizing E2 A and B conformations were isolated from infected humans, this indicates that both conformations exist during an infection and are accessible on the virus for neutralization. Thus, they provide important high-resolution structural information on HCV vulnerabilities to guide design of HCV vaccine antigens. How these conformations are related to conformational changes of Env during HCV entry to host cells remains to be determined.

## 3. Structure-Based and Germline-Targeting Vaccine Design

Structural studies of HCV Env and bnAbs, supplemented with biochemical and genetic data (e.g., NGS studies of antibody repertoires and bnAb developmental pathways), provide important insights into how bnAbs recognize and neutralize HCV and how they are generated. This knowledge will be crucial for the “reverse” engineering of vaccine antigens to elicit bnAbs. Over the past decade, several vaccine-design strategies have been proposed and evaluated for the induction of HCV bnAbs. These strategies include: (a) removal and suppression of immunodominant decoy epitope (see Section 2.3.3) [136,138,139,140]; (b) grafting of neutralizing epitopes onto heterologous scaffolds [144,145,146,147]; (c) stabilizing antigen structures [144,148,149,150]; and (d) presenting neutralizing epitopes on nanoparticles [32,144,151].

Several peptide-based antigens were engineered based on the complex structures of bnAbs and E2 linear neutralization epitopes (AS412 or AS434). Based on the structures of AS412 β-hairpin conformation, cyclic AS412 (cycAS412) antigens were designed in two independent studies. In the first study, mice-immunization experiments using cycAS412 resulted in the elicitation of high-affinity-binding Abs, but failed to elicit nAbs [152]. The structure of one of the isolated mAbs in complex with cycAS412 revealed that the mAb binds to the opposite face of the epitope, and the side chains of the critical residues for bnAb binding point to directions different from the neutralizing AS412 conformation. In the second study, cycAS412 peptides were designed based on a cyclic defensin protein and a bivalent E2 immunogen with two copies of the cyclic epitope fused on its surface [147]. Mice-immunization experiments resulted in greater AS412-specific binding and neutralization responses than the native peptide epitope, albeit with limited neutralization breadth [147]. An alternative strategy for the design of an AS412-mimic antigen to elicit AS412-like bnAbs was recently reported [144]. In this study, the bnAb AP33 (Table 1) was used as a template to create a structural mimic of its paratope to elicit anti-idiotype Abs. Biochemical and structural studies indicated that one of the anti-idiotype Abs, named B2.1A, mimics the AP33 epitope. Mice-immunization experiments with B2.1 A induced antibodies that can recognize the same epitope as AP33 and are able to protect against HCV challenges in such experiments.

The designing of a fully synthetic peptide-based antigen, using an epitope mimicry strategy, was recently reported [146]. Cyclic epitopes were chemically synthesized based on AS412, AS434, and epitope III. Next, the three cyclic epitopes were chemically conjugated to obtain a mimic of E2 discontinuous epitope and scaffolded on keyhole limpet hemocyanin (KLH) protein. Mice-immunization experiments succeeded in elicitation of E2-binding antibody response that failed to neutralize HCV, indicating further optimization of this epitope mimicry strategy is needed.

Recently, we designed an E2 antigen based on E2c3 [76,119], named E2mc3, by optimizing VR2 and the β-sandwich loop by a computational approach [136,141]. In E2 core structures, the VR2 is in close proximity to AR3, which may obscure neutralizing epitopes. Thus, truncation of VR2, as in E2c3 [76,119] and E2Δ123 (E2 with the truncation of HVR1, VR2, and VR3) [91], should eliminate this potential issue. In addition to the removal of the immunodominant decoy epitopes (HVR1) and variable loops (VR2 and VR3) in E2c3, ensemble-based de novo designing of a truncated VR2 resulted in improved stability and antigenicity of E2 without altering its key neutralizing epitopes as indicated by the crystal structures of the optimized E2–bnAbs complexes [136]. Displaying this optimized E2 antigen on self-assembly nanoparticles resulted in enhanced antigenicity and efficient elicitation of nAb responses in mice, underscoring the usefulness of structure-based design approach for the improvement of traditional vaccine antigens [136].

In addition to structure-based design, an emerging vaccine design approach is to target B cells that have the potential to develop bnAbs during vaccination [153,154]. Multidonor-class or public antibody response is a “germline-endowed” antibody response with shared genetic background and modes of binding identified in the immune responses against a given antigen in multiple infected or vaccinated individuals. Such a response has been described against different pathogens, including influenza virus [148,155,156], HIV-1 [149,150], dengue [151], SARS-CoV-2 [157], and malaria [158]. For HCV, the broad neutralization response is biased by V_H_1-69-encoded bnAbs that target AR3 [134]. V_H_1-69-encoded bnAbs exhibit a relatively low level of SHM, and for some of them, germline-reverted precursors can bind E2 with good affinity [118,119]. The neutralization breadth of these antibodies is acquired by SHM and affinity maturation [119]. These results suggest the potential of vaccine strategies to amplify such public antibody responses through targeting the corresponding antibody germline precursors to elicit broad neutralizing response. To evaluate this important antibody response by vaccination, it will require testing of the vaccine candidates in humans, closely related species or transgenic animals. We have recently investigated the relevance of animal models for vaccine development by dissecting the antibody responses to vaccination with the Chiron recombinant E1E2 complex in healthy volunteers, RMs, and mice [125]. RMs provide an attractive preclinical nonhuman primate (NHP) model for evaluation of human vaccine candidates because of their genetic relatedness to humans [159], with the RM and human VH genes sharing 92% homology on average [160]. The results demonstrated that NHPs, but not mice, are capable of faithfully recapitulating human antibody responses to the same E1E2 antigen. The majority of the nAbs were strain-specific; however, bnAbs against AR3 were also isolated from different animals. Genetic analysis revealed that most of the RM bnAbs were derived from VH1.36, an ortholog of the human V_H_1-69 gene (sharing 90% amino acid identity) [125,133]. Structural analysis of two RM bnAbs (RM2-01 and RM11-43) in complex with the E2c domain indicated that the RM bnAbs engaged AR3 using similar molecular features as for the human V_H_1-69-encoded public bnAbs, but using a different binding mode [133]. This study points to the usefulness of the RM model in HCV vaccine research.

## 4. Conclusions 

The development of highly effective DAAs against HCV raised the hope for HCV eradication. In 2016, the World Health Organization announced a global hepatitis strategy to reduce new hepatitis infections by 90% and deaths by 65% by 2030. The continuous increase in the number of new HCV infections highlights the real-world challenges in combating a human infection without a vaccine. A broadly effective vaccine remains a critical tool for the global control of HCV. The failure of a T-cell-based HCV vaccine demands an honest assessment of the vaccine strategies in the past, and alternative approaches. Recent advances in structural and genetic understanding of HCV Env and bnAb responses create new hope for the rational design of HCV vaccine antigens to elicit potent cross-nAbs in vaccination.

## Figures and Tables

**Figure 1 viruses-13-00833-f001:**
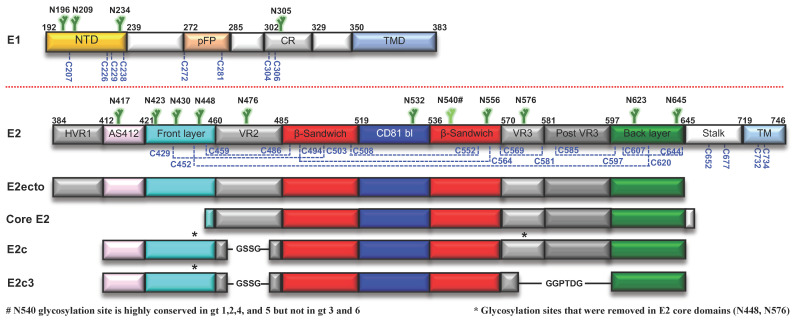
HCV E1 and E2 Env proteins. Schematic representation of the HCV E1 and E2 Env proteins and E2 core domains that were used for E2 structural studies, colored by structural component. Numbering is based on the H77 prototypic strain. N-linked glycans are shown in green, and the conserved cysteines in blue. For E2, disulfide bonds, based on the H77 E2C-AR3C structure, are shown in blue dashed lines. NTD, N’ terminal domain; pFP, putative fusion peptide; CR, conserved region; TMD, transmembrane domain; HVR1, hypervariable region 1; AS412, antigenic site 412; VR, variable region; CD81 bl, CD81 binding loop; TM, transmembrane.

**Figure 2 viruses-13-00833-f002:**
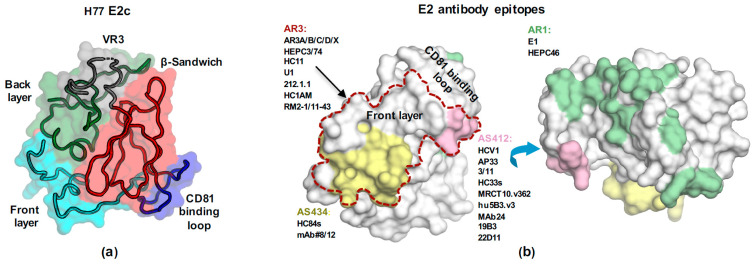
The structure of E2 Env. (**a**) The structure of H77 E2c (PDB: 4MWF) with the molecular surface color-coded as in Figure 1. The structures of E2 core domain reveal a novel protein fold consisting of a central immunoglobulin (Ig) β-sandwich fold (red), which is stabilized by the conserved disulfide bonds and flanked by an N-terminal front layer (cyan) and a C-terminal back layer (green). The long CD81 binding loop (blue) extends from the central β-sandwich domain. (**b**) E2 neutralization antigenic sites mapped onto the H77 E2c surface. The AR1 immune decoy site is also shown. AR3 is marked in dashed line, and AS412, AS434, and AR1 are colored in pink, yellow, and green, respectively. The mAbs that were used for structural studies and epitope characterization are listed.

**Figure 3 viruses-13-00833-f003:**
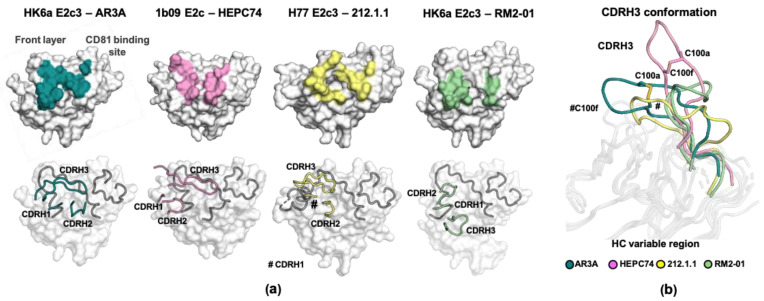
The binding mode of AR3-targeted V_H_1-69-encoded bnAbs. (**a**) Structural comparison of the epitopes and the CDR positions of the bnAb subgroups. AR3A is representative of the AR3A/B/C/D, U1 and HC11 group; HEPC74 of the HEPC74 and HEPC3 group; 212.1.1 of the 212.1.1 and HC1AM group; and RM2-1 of the RM2-1 and RM11-43 group. Top: the epitopes of AR3-targeted V_H_1-69-encoded bnAbs. E2c structures are shown in surface representation and antibody footprints are colored and labelled. Bottom: the position of the HC CDRs in the E2c–Fab structures. E2c domains are shown in surface representation with the front layer and CD81 binding loop as an illustration and colored in gray. The HC CDRs are also illustrated. (**b**) Conformation of CDRH3 of AR3A, HEPC74, 212.1.1, and RM2-01 from the E2–bnAbs complexes. The intra disulfide bond in the CDRH3 of AR3A and HEPC74 is shown in stick representation.

**Figure 4 viruses-13-00833-f004:**
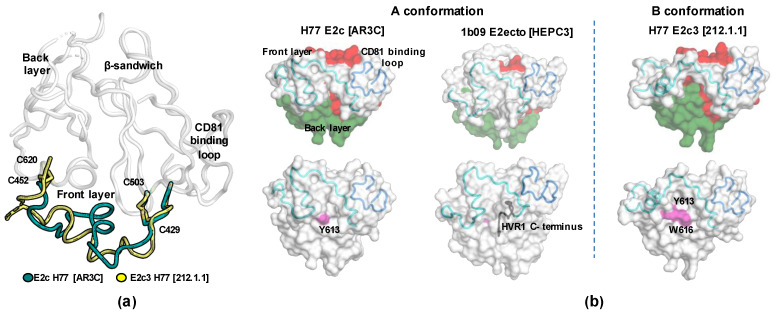
The conformations of E2 front layer. (**a**) Superposition of E2c from the crystal structures with AR3C and 212.1.1 indicating conformational changes of the front layer. The two disulfide bonds (C429-C503 and C452-C620) that represent the boundaries of the conformational changes are labeled. (**b**) Surface representation of E2c from the complex with AR3C (left) and 212.1.1 (right), and E2ecto from the complex with HEPC3 (middle). The conformational changes in the front layer of E2c3 in the 212.1.1 complex expose more of the β-sandwich and back-layer surfaces (colored in red and green respectively, upper panels) that include residues critical for CD81 binding (bottom panels, colored in pink).

**Table 1 viruses-13-00833-t001:** Summary of the structures of mAb Fab fragments in complex with epitope-derived linear peptides of HCV neutralization sites.

Epitope	mAbs		Antigen Conformation	PDB ID	References
AS412	HCV1	Human bnAb	β-hairpin	4DGV/Y	[94]
	AP33	Mouse bnAb	β-hairpin	4G6A	[93,97]
				4GAG/J	
	MRCT10.v362	Humanized and affinity-matured bnAb	β-hairpin	4HS6	[90]
	Hu5B3.V3	Humanized and affinity-matured bnAb	-hairpin	4HS8	[90]
	3/11	Rat bnAb	open	4WHY/T	[100]
	HC33.1	Human bnAb	semiopen	4XVJ	[98]
	HC33.4	Human bnAb	semiopen	5FGB	[99]
	HC33.8	Human bnAb	semiopen	5FGC	[99]
	MAb24	Mouse bnAb	β-hairpin	5VXR	[96]
	19B3	Mouse bnAb	β-hairpin	6BZU	[95]
	22D11	Mouse bnAb	β-hairpin	6BZY	[95]
	19B3 GL	Mouse bnAb	β-hairpin	6BZW	[95]
	22D11 GL	Mouse bnAb	β-hairpin	6BZV	[95]
AS434	HC84.1	Human bnAb	1.5-turn α-helix	4JZN	[102]
	HC84.27	Human bnAb	1.5-turn α-helix	4JZO	[102]
	HC84.26	Human bnAb	1.5-turn α-helix	5ERW	[103]
	HC84.26 AM	Affinity-matured human bnAb	1.5-turn α-helix	4Z0X	[103]
	8	Murine-neutralizing mAb (genotype 1a only)	1.5-turn α-helix	4HZL	[104]
	12	Murine-non-neutralizing mAb	1.5-turn α-helix	4Q0X	[105]
a.a. 529–540	DAO5	Non-neutralization mAb	one-turn α-helix	5NPH/I/J	[106]

**Table 2 viruses-13-00833-t002:** Summary of the structures and of the binding characterization of E2 core domains in complex with a V_H_1-69-encoded, AR3-targeting bnAbs Fab fragment.

E2 Domain	HCV Isolate ^1^	mAbs		Front Layer ^2^	CDRH2 Tip Sequence ^3^	CDRH3 Length ^4^		CxGGxC Motif		HC Dominancy	PDB ID	REF
E2c3	HK6a (6a)	AR3A	Human bnAb from chronically infected	A	FIPMF	18		√	HC only		6BKB, 6UYG	[119]
E2c3, E2mc3	HK6a (6a)	AR3B	Human bnAb from chronically infected	A	IIPAF	19			HC only		6BKC, 6UYF	[119,136]
E2c, E2mc3	H77 (1a)	AR3C	Human bnAb from chronically infected	A	VVPLF	18		√	HC only		4MWF, 6UYD/M	[44,136]
E2c3	HK6a (6a)	AR3D	Human bnAb from chronically infected	A	IIPFF	22			HC only		6BKD	[119]
E2c3	HK6a (6a)	U1	Human bnAb from chronically infected	A	ITPIF	17			HC dominant		6WO3	[121]
E2c3	HK6a (6a)	HC11	Human bnAb from chronically infected	A	IIPMF	17		√	HC dominant		6WO4	[121]
E2ecto ^5^	1b09 (1b), 1a53 (1a)	HEPC3	Human bnAb from spontaneously cleared patient	A	ITPIF	17		√	HC only		6MEI/J/K	[118]
E2ecto, E2c	1b09 (1b)	HEPC74	Human bnAb from spontaneously cleared patient	A	MSPIS	18		√	HC only		6MEH	[118]
E2c3 ^5^	H77 (1a)	HC1AM	Human bnAb from chronically infected	B	FIPMF	15			HC dominant		6WOQ	[121]
E2c3 ^5^	H77 (1a)	212.1.1	Human nAb from spontaneously cleared patient	B	SIPIL	16			HC dominant		6WO5	[121]
E2c3	HK6a (6a)	RM2-01	bnAb from immunized rhesus macaques	A	IVPLG	15			HC only		7JTF	
E2c3	HK6a (6a)	RM11-43	bnAb from immunized rhesus macaques	A	IIPLG	19			HC only		7JTG	
E2ecto	1b09 (1b)	AR3X	Human bnAb from chronically infected	A	INPIS	19		√			6URH	[120]
Core E2 ^6^	J6	2A12	Non-neutralization mAb								4WEB	[106]

^1^ The virus genotype is indicated in brackets. ^2^ The conformation of the front layer. ^3^ Sequence of CDRH2, amino acid 51–54. The highly conserved germline encoded P52a is highlighted in bold. ^4^ CDRH3 length (amino acid). ^5^ These complex structures also contain a non-nAb (Fab E1 for HC1AM and 212.1.1, and HEPC46 for HEPC3) to enhance the crystallization. ^6^ 2A12 in a non-nAb that binds to E2 back layer region. 2A12 is not encoded by V_H_1-69 genes.

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
