# Peer review of "From Structural Studies to HCV Vaccine Design"

_viruses, 2021, doi:10.3390/v13050833_

Round 1

Reviewer 1 Report

The review by Yechezkel et al aims to provide an overview of structural studies of the E1 and E2 envelope proteins of hepatitis C virus and their potential implication in HCV vaccine design.

The manuscript is well written and pleasant to read considering the complexity of the subject.  It is well documented, very complete and provides an excellent synthesis of the literature on the subject. The biography is up to date and takes into consideration the latest work published on the subject.  In addition, the subject matter makes an important contribution to the field, as, although benefiting from a very effective treatment, infection with the hepatitis C virus is still a major health problem. 

I have only a few minor comments:

- a short paragraph introducing the various epitopes detailed later would be much appreciated at the beginning of chapter 3.3

- the paragraph starting at line 307 on page 8 about epitope III should be a separate subchapter of chapter 3.3.  In this paragraph, the authors should better clarify that epitope III includes the CD81-binding loop and a portion of the core Ig-like domain

- the paragraph from line 341 to 348 page 8 gives some information that is redundant with chapter 3.4.  The authors should merge this paragraph with chapter 3.4, which should also be somewhat reorganized and shortened.

- paragraph 3.7 is extremely dense and would gain in readability with the addition of a table summarizing all the vaccine designs discussed.

-the recent review by Sepulveda-Crespo et al (2020) about Hepatitis C virus vaccine with a focus on humoral immune response should be included in the manuscript

Author Response

The review by Yechezkel et al aims to provide an overview of structural studies of the E1 and E2 envelope proteins of hepatitis C virus and their potential implication in HCV vaccine design.

The manuscript is well written and pleasant to read considering the complexity of the subject.  It is well documented, very complete and provides an excellent synthesis of the literature on the subject. The biography is up to date and takes into consideration the latest work published on the subject.  In addition, the subject matter makes an important contribution to the field, as, although benefiting from a very effective treatment, infection with the hepatitis C virus is still a major health problem. 

I have only a few minor comments:

- a short paragraph introducing the various epitopes detailed later would be much appreciated at the beginning of chapter 3.3

Response: We thank the reviewer for the positive and supportive review and suggestions on improving our review. A short paragraph was added to the introduction of the “structural studies of E2” chapter (now chapter 2.3.1 lines 236-241). 

- the paragraph starting at line 307 on page 8 about epitope III should be a separate subchapter of chapter 3.3.  In this paragraph, the authors should better clarify that epitope III includes the CD81-binding loop and a portion of the core Ig-like domain

Response: The paragraph about epitope III is now a separated subchapter (2.3.1.3, lines 322-331).  We also clarified that the epitope is composed of the C-terminus of the CD81 binding loop and the β-strand 6 of the β-sandwich scaffold region.

- the paragraph from line 341 to 348 page 8 gives some information that is redundant with chapter 3.4.  The authors should merge this paragraph with chapter 3.4, which should also be somewhat reorganized and shortened.

Response: Based on the reviewer's comment, we merged the paragraph about the genetic analysis of AR3 bnAbs into the chapter about the binding mode of AR3-targeting bnAbs (now chapter 2.3.2, line 389). In chapter 2.3.2, we summarized recently published structural data that indicated that there are different modes of binding to AR3 that change the prevailing opinion about the binding of AR3-like bnAbs to AR3. Therefore, we decided to write this chapter in more detail. 

- paragraph 3.7 is extremely dense and would gain in readability with the addition of a table summarizing all the vaccine designs discussed. the recent review by Sepulveda-Crespo et al (2020) about Hepatitis C virus vaccine with a focus on humoral immune response should be included in the manuscript

 Response: We thank the reviewer for this commend, we added the reference to the review (lines 66 and 288).

Reviewer 2 Report

The authors wrote a very comprehensive and profound review article about structural studies of HCV envelope glycoproteins and about epitopes for neutralizing antibodies. This knowledge was translated into different rational vaccine design strategies which are summarized in this manuscript.

The manuscript is very well written and citations are appropriate. The manuscript would benefit from a more comprehensible structure, details are given below.

  • Section 2 should be integrated in section 3 to avoid jumps forth and back. Below three examples are given:
    • E1 function and structure is explained in line 131 -146 and again in section 3.2. (line 224-line 234).
    • Description of E1E2 heterodimer organization and E1E2 epitopes in line 124 -130 would perfectly fit to section 3.1. (line 193 -223).
    • Furthermore, E2 structure is partly described in section 2 (line 154 -172) and partly in section 3 (line 235 -377).

The following slightly changed structure should be considered:

  1. Introduction
  2. HCV Env glycoprotein
    • E1E2 Heterodimer
    • E1 Glycoprotein
    • E2 Glycoprotein
      • Structural studies of HCV Env glycoproteins
        • AS412
        • AS434
        • E2 core structure
      • The binding mode of AR3-targeting bnAbs
      • E2 immunodomiant decoy epitopes
      • Structural plasticity of E2
  1. Structure-based and germline-targeting vaccine design
  2. Conlcusion
  • The first passage in section 3 (line 173 -183) is about structural plasticity of E2 and should be moved to the section 3.6. (line 470 -502) which focuses on this topic.

Furthermore, some specific points should be addressed to improve the quality of the manuscript:

Line 146: The conserved region of E1 is targeted not only by th nAB IGH526 but also by IGH505.

Line 182 -183: Reference is missing to prove the statement that “ectodomain of E2 can be expressed as a folded and soluble monomeric protein, and thus is amenable for structural studies.”

Figure 1, Legend: Appreviation for VR, variable region is missing

Table 1, title: Three HCV neutralization sites are shown in the table but in the title it is stated that “two HCV neutralization sites” are shown. The title should be changed accordingly.

Footnotes of table 2 are wrongly allocated in line 461 -469

The non-nAb DAO5 (Vasiliauskaite et al., Conformational Flexibility in the Immunoglobulin-Like Domain of the Hepatitis C Virus Glycoprotein E2, mBio, 2017) should be integrated in this review to give a broader overview on conformational flexibility in the CD81-binding loop.

Author Response

Reviewer 2:

The authors wrote a very comprehensive and profound review article about structural studies of HCV envelope glycoproteins and about epitopes for neutralizing antibodies. This knowledge was translated into different rational vaccine design strategies which are summarized in this manuscript.

The manuscript is very well written and citations are appropriate. The manuscript would benefit from a more comprehensible structure, details are given below.

  • Section 2 should be integrated in section 3 to avoid jumps forth and back. Below three examples are given:
    • E1 function and structure is explained in line 131 -146 and again in section 3.2. (line 224-line 234).
    • Description of E1E2 heterodimer organization and E1E2 epitopes in line 124 -130 would perfectly fit to section 3.1. (line 193 -223).
    • Furthermore, E2 structure is partly described in section 2 (line 154 -172) and partly in section 3 (line 235 -377).

The following slightly changed structure should be considered:

  1. Introduction
  2. HCV Env glycoprotein
    • E1E2 Heterodimer
    • E1 Glycoprotein
    • E2 Glycoprotein
      • Structural studies of HCV E2 Env glycoproteins
        • AS412
        • AS434
        • E2 core structure
      • The binding mode of AR3-targeting bnAbs
      • E2 immunodomiant decoy epitopes
      • Structural plasticity of E2
  1. Structure-based and germline-targeting vaccine design
  2. Conlcusion
  • The first passage in section 3 (line 173 -183) is about structural plasticity of E2 and should be moved to the section 3.6. (line 470 -502) which focuses on this topic.

Response: We thank the reviewer for the positive and supportive review and for the suggestions regards the improvement of the review structure. The review was reorganized based on the reviewer’s suggestion.

Furthermore, some specific points should be addressed to improve the quality of the manuscript:

Line 146: The conserved region of E1 is targeted not only by th nAB IGH526 but also by IGH505.

Response: Thank you for the comment, we add the non-nAb IGH505 (line 185).

Line 182 -183: Reference is missing to prove the statement that “ectodomain of E2 can be expressed as a folded and soluble monomeric protein, and thus is amenable for structural studies.”

Response: We add references (line 135).

Figure 1, Legend: Abbreviation for VR, variable region is missing

Response: Thank you for the comment, we add the abbreviation for VR to the legend of figure 1.

Table 1, title: Three HCV neutralization sites are shown in the table but in the title, it is stated that “two HCV neutralization sites” are shown. The title should be changed accordingly.

Response: We agree with the reviewer that the table summarizes the structures of Fabs with linear peptides of three sites, two are neutralization site and the third is the non-neutralizing site 529-540. Therefore, we changed the title to “Summary of the structures of mAb Fab fragments in complex with epitope-derived linear peptides of HCV neutralization sites”

Footnotes of table 2 are wrongly allocated in line 461-469.

Response: Thank you, the footnote location was corrected.

The non-nAb DAO5 (Vasiliauskaite et al., Conformational Flexibility in the Immunoglobulin-Like Domain of the Hepatitis C Virus Glycoprotein E2, mBio, 2017) should be integrated in this review to give a broader overview on conformational flexibility in the CD81-binding loop.

Response: The non-nAbs DAO5 is included in the review (lines 322-331 and table 1)

Reviewer 3 Report

 Yechezkel and colleagues presented an interesting review about the recent advances in structural studies of HCV Envelope (Enc) and Env-neutralizing antibody complexes and how this impact HCV vaccine design. The review includes HCV Env glycoproteins (HCV E1 and E2 Env proteins), Structural studies of HCV Env glycoproteins including structural studies of E1E2, structural studies of E1, and structural studies of E2. The authors also showed a summary of the structures of mAb Fab fragments in complex with epitope-derived linear peptides of two HCV neutralization sites. Also the authors summerized the structures and of the binding characterization of E2 core domains in complex with VH1-69 en- coded AR3-targeting bnAbs Fab fragment.  Furthermore, the authors discussed E2 immunodominant decoy epitopes and E2 plasticity. Finally, the authors showed how this stuctural activities of HCV Env glycoproteins affect the the HCV vaccine design.

In general, the manuscript is well designed, nicely written, has an acceptable flow, comperhensive, and desreved for publication.

Author Response

Yechezkel and colleagues presented an interesting review about the recent advances in structural studies of HCV Envelope (Enc) and Env-neutralizing antibody complexes and how this impact HCV vaccine design. The review includes HCV Env glycoproteins (HCV E1 and E2 Env proteins), Structural studies of HCV Env glycoproteins including structural studies of E1E2, structural studies of E1, and structural studies of E2. The authors also showed a summary of the structures of mAb Fab fragments in complex with epitope-derived linear peptides of two HCV neutralization sites. Also the authors summarized the structures and of the binding characterization of E2 core domains in complex with VH1-69 en- coded AR3-targeting bnAbs Fab fragment.  Furthermore, the authors discussed E2 immunodominant decoy epitopes and E2 plasticity. Finally, the authors showed how this stuctural activities of HCV Env glycoproteins affect the the HCV vaccine design.

In general, the manuscript is well designed, nicely written, has an acceptable flow, comperhensive, and desreved for publication.

Response:

We thank the reviewer for the supportive review